# The Influence of Male Ejaculatory Abstinence Time on Pregnancy Rate, Live Birth Rate and DNA Fragmentation: A Systematic Review

**DOI:** 10.3390/jcm12062219

**Published:** 2023-03-13

**Authors:** Freja Sørensen, Linda Magnusson Melsen, Jens Fedder, Sinor Soltanizadeh

**Affiliations:** 1Centre of Andrology, Fertility Clinic, Department D, Odense University Hospital, DK-5000 Odense, Denmark; 2Department of Obstetrics and Gynecology, Fertility Clinic, Copenhagen University Hospital, DK-2730 Herlev, Denmark; 3Department of Clinical Medicine, University of Southern Demark, DK-5000 Odense, Denmark; 4Gynaecological Department, Zealand University Hospital, DK-4000 Roskilde, Denmark

**Keywords:** ejaculatory abstinence, pregnancy rate, live birth rate, DNA-fragmentation, infertility

## Abstract

Variation in ejaculatory abstinence time and its influence on semen quality and clinical reproductive outcomes is a growing concern among clinicians and researchers. The WHO (World Health Organization) recommends 2–7 days of abstinence time prior to semen collection for diagnostic purposes; however, the evidence that such an abstinence period leads to better pregnancy outcomes remains unclear. The aim of this systematic review is to evaluate short and long ejaculatory abstinence time in association with pregnancy rate, live birth rate and DNA fragmentation, in order to make a recommendation on an ideal timeframe for ejaculatory abstinence. This review is conducted according to the PRISMA guidelines and registered in PROSPERO (CRD42022379039). The electronic databases PubMed, Embase and Cochrane were searched for eligible studies. The Scottish Intercollegiate Guidelines Network was used for the assessment of the risk of bias across the included studies. Twenty-four studies were included in this systematic review. The included studies confirm that a shorter abstinence time is associated with improved pregnancy rates and live birth rates following assisted reproductive technology compared with longer ejaculatory abstinence times at different cut-off points. Similarly, a lower DNA fragmentation index was reported in semen analyses collected from short abstinence times compared with long abstinence times. However, due to the heterogeneity of the included studies, it is not possible to extract an ideal time of ejaculatory abstinence, but all outcomes improved with shorter ejaculatory abstinence times. This systematic review confirms that short ejaculatory abstinence times, less than those recommended by the WHO for diagnostic purposes, are associated with higher pregnancy and live birth rates and improved DNA fragmentation, when compared to long ejaculatory abstinence times.

## 1. Introduction

Approximately 24% of couples of reproductive age are primary or secondary infertile, which means they have not been able to conceive after one year of unprotected sexual intercourse. Male factor infertility affects over 50% of these infertile couples, either in isolation or in combination with a female factor [1]. Male factor infertility is usually defined by abnormal results on semen analysis according to the World Health Organization (WHO) [2].

It is well known that several factors can affect semen quality, including general health status, metabolic syndrome, osteoporosis, age, medication, body mass index and lifestyle including diet, smoking, caffeine and alcohol intake [3,4]. Semen parameters can vary both between and within individuals, and one factor causing intraindividual variation is suggested to be ejaculatory abstinence time (EA) [5]. When spermatozoa pass the epididymal tract, they undergo a series of biochemical and physiological changes to mature them into a fertilizing component. During this transit, spermatozoa may be exposed to reactive oxygen species (ROS) [6,7]. ROS is known to cause DNA damage and fragmentation, which can negatively affect fertility [8]. Studies are suggesting that DNA fragmentation is associated with poor pregnancy and live birth rates following assisted reproductive technology (ART) [9,10]. Testicular sperm has a lower DNA fragmentation when compared to ejaculated sperm, which may indicate that DNA fragmentation is increased during the transition and storage in the epididymal duct [11]. Therefore, EA may influence DNA fragmentation and clinical reproductive outcomes.

The sixth edition of the WHO manual for examining and processing human semen has been recently released, recommending 2–7 days of EA prior to semen collection for diagnostic purposes [2]. Several recent studies have reported a correlation between short EA and lower DNA fragmentation and thus better semen quality [12,13]. Much indicates that a short EA in the context of ART may lead to higher pregnancy rates. A recent meta-analysis found significantly higher pregnancy rates when the EA was less than 4 days compared with 4 to 7 days; however, the meta-analysis only included four studies, which is why an updated, broader search is needed to strengthen the evidence [14]. Also, the European Society of Human Reproduction and Embryology (ESHRE) is currently recommending a shorter and narrower range from 3 to 4 days of EA [15].

If a shorter EA can have a positive impact on pregnancy outcomes, it is a safe, low-cost and non-invasive intervention in fertility treatment. Currently, there is no recent systematic review focusing on pregnancy rate, and the available recent reviews only include a few studies.

The aim of this systematic review is to compare short and long ejaculatory abstinence time and investigate the association between fertility outcomes and semen quality. The main objective was pregnancy rate and secondary objectives were live birth rate and DNA fragmentation index.

## 2. Materials and Methods

This systematic review was conducted and reported in accordance with Preferred Reporting Items for Systematic Reviews and Meta-Analysis (PRISMA) guidelines [16] and was submitted to PROSPERO, date 30 November 2022, and accepted 11 December 2022. The registration number was CRD42022379039.

### 2.1. Eligibility Criteria

A study was found eligible for inclusion if the following inclusion criteria were met:P: The study population was men of reproductive age, including men referred to fertility treatment.I: The intervention was a short abstinence time between successive ejaculation.C: The comparison was a long abstinence time between successive ejaculation.O: The main outcome was pregnancy rate, and secondary outcomes were live birth rate and DNA fragmentation.

All types of original published human studies were included, while expert opinions, case studies and protocols were excluded. All types of fertility treatment were included. Publication languages was limited to English, Spanish and Scandinavian languages.

### 2.2. Information Sources and Search Strategy

A research librarian conducted a systematic search in PubMed, Embase and Cochrane in April 2022. An initial search strategy was performed with the following search string: *((male or paternal or man or oligospermia or Oligozoospermia or low sperm count or semen or sperm or Asthenospermia or Asthenozoospermia or Azoospermia or Teratozoospermia or Teratospermia or Spermatogenic dysfunction) and (ejaculation or ejaculatory) and (abstinence or time or interval) and (pregnancy or pregnant or live birth or hCG or human chorionic gonadotropin or DNA fragmentation)).* All suggested MeSH terms were included in the search on PubMed. Additional articles were manually retrieved after reviewing the reference lists from relevant publications.

### 2.3. Study Selection and Data Collection

All potentially eligible studies were independently screened by two investigators (F.S. and L.M.) based on the title and abstract, and potentially eligible studies were subsequently full-text screened for inclusion. Covidence.org was used as the screening tool. Disagreements between reviewers were resolved by a third person (S.S.). One review author extracted data from the included studies and the second author independently and systematically cross-checked all the data. Disagreements were resolved by discussion between the two review authors (F.S. and L.M.).

### 2.4. Data Items

The study demographics, study country, year of publication, study design, population size, subjects, cause of infertility, investigated abstinence time and primary study outcomes were extracted. Data regarding pregnancy rate, live birth rate and the DNA fragmentation index were extracted along with the type of ART and assay method for DNA fragmentation analysis. Missing data were sought to be collected by contacting corresponding authors of reference publications.

### 2.5. Risk of Bias Assessment

To assess the quality of evidence of the included cohort studies, two investigators (L.M. and F.S.) independently evaluated the quality according to the Scottish Intercollegiate Guidelines Network (SIGN). SIGN is a methodology checklist for cohort studies, to evaluate the risk of bias according to the selection, assessment, confounding, and statistical analysis [17]. We excluded irrelevant questions from our study, such as 1.3 and 1.4 in the selection section, which were not applicable since EA would not be affected by the participation rate or outcomes prior to enrollment. In the assessment section, questions 1.8, 1.9 and 1.12 were also not applicable, as blinding EA was not possible. Disagreements between reviewers were resolved by discussion with a third author (S.S.).

### 2.6. Data Synthesis

For data presentation of the included studies, the following syntheses were made:(1)Quantitative analysis: The proportion of studies that reported pregnancy rate, live birth rate and DNA fragmentation were compared for subjects with short versus long EA. The results of studies performing statistical analyses are similarly presented in a separate table.(2)Visual analysis: Studies reporting pregnancy rates as percentages were illustrated in a graph grouped by EA on the x-axis. In the studies where a time interval of abstinence days was used, the mean day of the interval was chosen for illustration.

## 3. Results

### 3.1. Study Selection

The search strategy is illustrated in the PRISMA flow diagram (Figure 1). A total of 1235 studies were identified after performing the search string across databases, and 1030 articles remained for primary screening after the removal of duplicates. We excluded 973 studies based on irrelevant titles and abstracts. Additionally, 20 studies were excluded during full-text review due to papers being unavailable or due to language restrictions. Furthermore, 15 studies were excluded as they did not meet the eligibility criteria. Twenty-two studies met the inclusion criteria for this systematic review. Additionally, two studies met the eligibility criteria when screening references for relevant papers. A total of 24 studies were included for further analysis, with a total of 14,173 cases.

The studies included in this review were conducted in Asia, North and South America and Europe. The largest studies were conducted in Mexico, Brazil, and India and included more than 9000 cases in total (Table 1). The majority of the studies focused on patients undergoing fertility treatment, while four of the studies included volunteers. The cause of referral to fertility treatment included male and female infertility, as well as mixed and unexplained infertility. The median age of the participants was between 30 and 40 years in the majority of the studies and ranged from 20 to 50 years. The EA varied from less than one hour to as long as 15–20 days. 

### 3.2. Level of Study Evidence

An overview of the studies’ risk of bias assessment is presented in Table 2. According to the SIGN methodology checklist, three of the included studies were of high quality [18,21,23], 17 had acceptable quality and four had low quality. Three of the studies with low quality had study populations under 11 [20,30,31], which increases the risk of Type II error due to low power and thus weakens the strength of evidence. Eight studies had large populations (*n* ≥ 800), of which six were retrospective and can according to the SIGN methodology checklist only maximally be rated as acceptable; however, three of these were of high quality in all other domains [36,38,40]. Six studies were downgraded from high quality to acceptable quality due to missing confidence intervals.

All the included studies had a well-executed assessment. Most of the studies address confounding as age, female factor, BMI, illness and smoking. For a detailed risk of bias assessment, see Appendix B.

**Table 2 jcm-12-02219-t002:** Overview of risk of bias assessment of the included studies.

Dahan et al. [18]	Kabukçu et al. [19]	Agarwal et al. [20]	Borges et al. [21]	Vahidi et al. [22]	Comar et al. [23]	Uppangala [24]	Sánchez-Martín et al. [25]	Scarselli et al. [26]	Shen et al. [27]	Gosálvez et al. [28]	Jurema et al. [29]	Mayorga-Torres et al. [30]	Jonge et al. [31]	Kulkarni et al. [32]	Marshburn et al. [33]	Barbagallo et al. [34]	Kably-Ambe et al. [35]	Gupta et al. [36]	Manna et al. [37]	Azizi et al. [38]	Welliver et al. [39]	Periyasamy et al. [40]	Lee et al. [41]	
																								Question
																								Selection
																								Assessment
							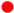																	Confounding
																								Statistical analysis
		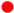					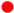					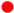	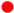											Quality


: low risk of bias, 

: acceptable risk of bias, 
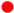
: high risk of bias.

### 3.3. Pregnancy Rate

As presented in Table 3, 13 studies examined the influence of varying abstinence time on pregnancy rate. Nine of the studies found a significantly higher pregnancy rate with short EA compared with long EA. The remaining four studies reported similarly higher pregnancy rates with shorter EA; however, they were statistically non-significant. Similarly, Figure 2 illustrates higher pregnancy rates with short EA compared with long EA, favoring better pregnancy outcomes of short EA. All 13 studies recruited patients from fertility clinics.

### 3.4. Live Birth Rate

Three studies evaluated live birth rates in relation to various abstinence times (Table 3). All three studies [27,34,40] found a significantly higher live birth rate when comparing short EA with long EA. One of the studies found significantly higher live birth rates with short EA of 1–3 h (65.2%) compared with 3–7 days (47.7%) in cryopreserved blastocysts (frozen thawed cycles), while results of the fresh cycle were non-significant but still reported higher live birth rates with short EA [27].

### 3.5. DNA Fragmentation Index

Sperm DNA fragmentation in relation to different EAs was reported in 15 studies (Table 3). Eleven studies found a significantly lower DNA fragmentation index (DFI) with short EA compared with long EA. The remaining four studies reported non-significant estimates, all with low DFI rates favoring short EA. However, three of the studies with non-significant results had populations less than 20 [30,31,39]. Some studies used interval days, successive days, or both. One study compared a group undergoing an EA of 2–7 days compared with a group receiving antioxidant therapy for three months and undergoing an EA of 3 days [22]. Three of the 15 studies investigated DFI rates in two independent groups; however, two of these studies had a large population (*n* = 818 and *n* = 2458). No significant differences were reported in the patient characteristics in the studies [21,23]. In the one study that did not have a large sample size (*n* = 120), no significant difference in DFI was found [19]. EAs of one day or less were associated with the lowest rates of DFI in the studies, indicating that sperm DNA quality may be worsened by longer EA.

For further information regarding statistical analyses on pregnancy rate, live birth rate and DFI, see Appendix A.

**Table 3 jcm-12-02219-t003:** Associations between ejaculatory abstinence time and pregnancy rate, live birth rate and DNA fragmentation.

Author	EA	ART	Pregnancy Rate	Live Birth Rate	DFI
Dahan et al. [18]	3 h and 3 days	ICSI, IVF			↓
Kabukçu et al. [19]	1 and 3 days	IUI	←→		←→
Agarwal et al. [20]	<2, 2–7 and 9–11 days 1, 2, 5, 7, 9, 11 days				↓
Borges et al. [21]	<4 and >4 days 1, 2, 3 and 4 days	ICSI	↑		↓
Vahidi et al. [22]	24 h, 3 and 2–7 days.				↓
Comar et al. [23]	<2, 2–5 and >5 days				↓
Uppangala et al. [24]	1, 3, 5 and 7 days				↓
Sánchez-Martín et al. [25]	12 h and 4 days	ICSI	↑		↓
Scarselli et al. [26]	1 h and 2–5 days	ICSI	←→		
Shen et al. [27]	1–3 h and 3–7 days	IVF	↑	↑	↓
Gosálvez et al. [28]	(1) 24 h and 4 days (2) 3 h and 4 days	ICSI			↓
Jurema et al. [29]	≤3, 3–10 and >10 days	IUI	↑		
Mayorga-Torres et al. [30]	1 and 3–4 days				←→
Jonge et al. [31]	1, 3, 5 and 8 days				←→
Kulkarni et al. [32]	1–3 h and 2–7 days				↓
Marshburn et al. [33]	<2, 3–5 and >5 days	IUI	↑		
Barbagallo et al. [34]	1 h and 2–7 days	ICSI	↑	↑	
Kably-Ambe et al. [35]	0–1, 2–3, 4–5, 6–7, 8–9, 10–14 and 15–20 days	IUI	↑		
Gupta et al. [36]	1, 2–5, 6–7 and ≥ 8 days	ICSI	↑		
Manna et al. [37]	1 h and 2–7 days	ICSI			↓
Azizi et al. [38]	1, 2, 3, 4, 5 and 6–10 days	ICSI	←→		
Welliver et al. [39]	1 and 3–5 days				←→
Periyasamy et al. [40]	2–4, 2–7, 5–7 and >7 days	ICSI or ICSI + IVF	↑	↑	
Lee et al. [41]	2–7 and 8 days	ICSI	←→		

↑: increase significantly with decreasing abstinence time (*p* < 0.05), ↓: decrease significantly with decreasing abstinence time (*p* < 0.05), ←→: not significantly different, EA: ejaculatory abstinence time, ICSI: intracytoplasmic sperm injection, IVF: in vitro fertilization, IUI: intrauterine insemination.

**Figure 2 jcm-12-02219-f002:**
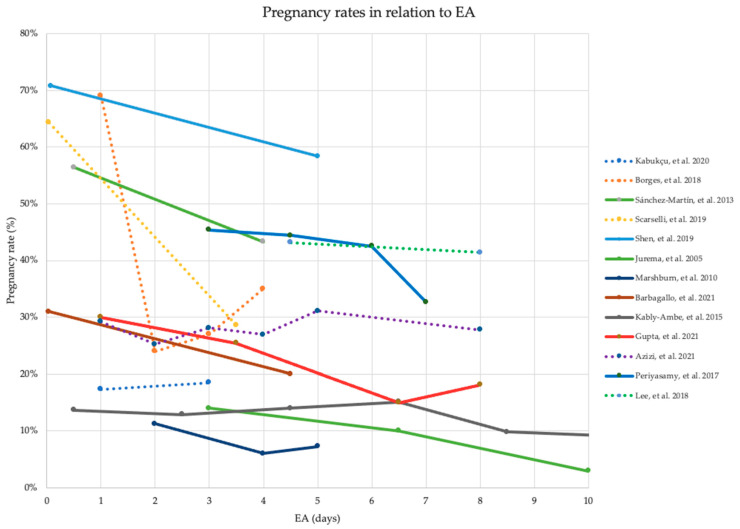
Pregnancy rates and ejaculatory abstinence time of 13 included studies reporting pregnancy rate. Solid line: statistically significant difference (*p* < 0.05), dotted line: no statistically significant difference (*p* > 0.05), EA: ejaculatory abstinence time [19,21,25,26,27,29,33,34,35,36,38,40,41].

## 4. Discussion

In the present review, 24 studies were included and evaluated for associations between men’s EA and reproductive outcomes as well as DNA fragmentation. According to the current evidence, pregnancy rates were higher with short EA compared with long EA. Also live birth rates were significantly higher with short EA in all the three studies reporting live birth rate compared with long EA. Additionally, DNA fragmentation was shown to be decreasing with shorter EA. Current evidence thus supports the beneficial outcomes of short EA regarding reproductive outcomes, and beneficial effects are seen in successive ejaculatory intervals as low as 1–3 h [27].

### 4.1. Strengths and Limitations

The strength of this systematic review was the meticulous and broad search strategy performed by a research librarian, which resulted in a large number of included studies. Additionally, the evaluation of all available studies including different abstinence intervals, populations and ART methods emphasizes the beneficial effects of short EA in all clinical settings regarding reproductive outcomes. Furthermore, this systematic review adhered to recommendations set by the PRISMA guideline [16].

There are some limitations in this systematic review. According to the SIGN risk of bias assessment tool, only three studies were of high quality, of which none of them reported live birth rates, while the rest of the studies were not rated higher than acceptable quality due to weaknesses in one or more domains. The inclusion of studies with small sample sizes leads to a small statistical power which might have increased the risk of Type II errors. Many of the included studies used different EA, leading to heterogenous data and it was therefore not possible to conduct a meta-analysis. Furthermore, different techniques to measure DFI were used in the studies e.g., sperm chromatin dispersion test and flowcytometry along with variations in cut-off values of DFI, which may impact the correlations between EA and DFI and limit the external validity of the results. It was not possible to assess possible confounders, such as lifestyle, smoking status, and daily coffee intake in the current evidence, although previous studies have found a significant correlation between these factors and semen quality and thus reproductive outcomes [42,43].

### 4.2. Comparison

This systematic review includes the highest amount of studies to date and finds supporting results when compared with three earlier systematic reviews investigating the influence of EA on semen quality and clinical outcomes following ART [12,13,44]. One review from 2021 compared <2, 2–7 and >7 days of EA and found the highest pregnancy and live birth rate with EA of <2 days compared with 2–7 and >7 days. Regarding DNA fragmentation, 11 out of 16 included studies additionally found comparable results with lower DFI when comparing <2 days of EA with 2–7 days [13]. The two other systematic reviews included three or fewer papers reporting pregnancy rate, and both found the highest pregnancy rate for intracytoplasmic sperm injection (ICSI) treatment after less than one day of EA, and under two [12] and three days [44] of EA for intrauterine insemination (IUI) treatment. In contrast to the previously mentioned reviews, our review included 13 studies evaluating pregnancy rates in relation to varying EAs. All previous systematic reviews and meta-analysis had five or fewer studies reporting pregnancy rate included. A potential explanation for this considerably high difference in the number of included articles might be the use of several databases and broader search strings, and in addition, the inclusion of recently published papers.

This systematic review focused mainly on pregnancy rates. A meta-analysis from 2020 did this as well, but only included four articles. The meta-analysis compared <4 days of EA with 4–7 days. Compared with long EA, short EA improved pregnancy rates significantly with an odds ratio of 1.44 (95%CI [1.17–1.78; *p* = 0.0006] in the forest plot analysis [14]. In the present review, most of the included studies reporting pregnancy rates found the highest rates with EAs of less than four days. Five studies showed significantly higher pregnancy rates with EA as short as one day or less when compared to longer abstinence times; for example, Gupta et al. [36] analyzed a large sample of 1691 cycles and found significantly higher pregnancy rates with one day of EA compared to 2–5, 6–7 and >8 days.

DNA fragmentation has been linked to impaired fertilization, suboptimal embryo quality, reduced pregnancy rates and increased spontaneous abortion rate after in vitro fertilization (IVF) treatment [10]. A systematic review and meta-analysis from 2015 found that a higher number of spermatozoa with DNA fragmentation in couples undergoing ART are associated with poorer outcomes [10]. In this systematic review, 11 studies found a significant correlation between DFI and EA, and all of them found that DFI rates were lower with shorter EA—especially EA of one day or less was associated with the lowest rates of DFI. Five studies had an EA of 3 h or less and all of them found significantly lower DFI when compared to a longer EA. A recent systematic review and meta-analysis from 2022 also found significantly lower DFI when comparing EA < 4 h with longer EA in a forest plot analysis of four studies [45]. This is supporting the theory that a short EA leads to lower DNA fragmentation, which may improve reproductive outcomes following ART.

Current evidence supports the theory that short EA is beneficial for reproductive outcomes; however, the risk of bias is still present. For conclusive and strengthened evidence, future studies are required in a prospective and randomized setting. For studies investigating pregnancy rate and live birth rate, we recommend a prospective randomized controlled trial in a large population undergoing similar ART with participants randomized to different successive ejaculatory abstinence intervals. This may establish the optimal interval of EA. For studies investigating DNA fragmentation, the same participant should be used as the control for their own samples for minimal confounding, eliminating interindividual variations in semen quality. Based on current evidence, this systematic review finds a clear trend that short EAs may enhance pregnancy rate, live birth rate and DFI. Revising the recommended 2–7 days of EA in the WHO manual regarding collection and processing of semen samples may lead to different reference intervals during diagnostic processing. Therefore, it may be worth considering only using an EA shorter when initiating fertility treatment. However, the ideal timeframe of EA can vary depending on the type of fertility treatment being used, as ideal EA’s may differ between fertilization provided by intercourse, IUI or IVF/ICSI, due to differences in semen volumes and sperm counts. These suggestions should be considered for future recommendations regarding EA in relation to fertility treatment.

## 5. Conclusions

Pregnancy rate, live birth rate and DNA fragmentation are likely to improve with short EA compared to long EA. Although it is not possible to conduct a clear recommendation on the ideal timeframe for an EA due to heterogenous abstinence times, this systematic review finds a clear tendency that a short EA is likely to improve pregnancy and live birth rate and decrease the level of DNA fragmentation in semen followed by ART.

## Figures and Tables

**Figure 1 jcm-12-02219-f001:**
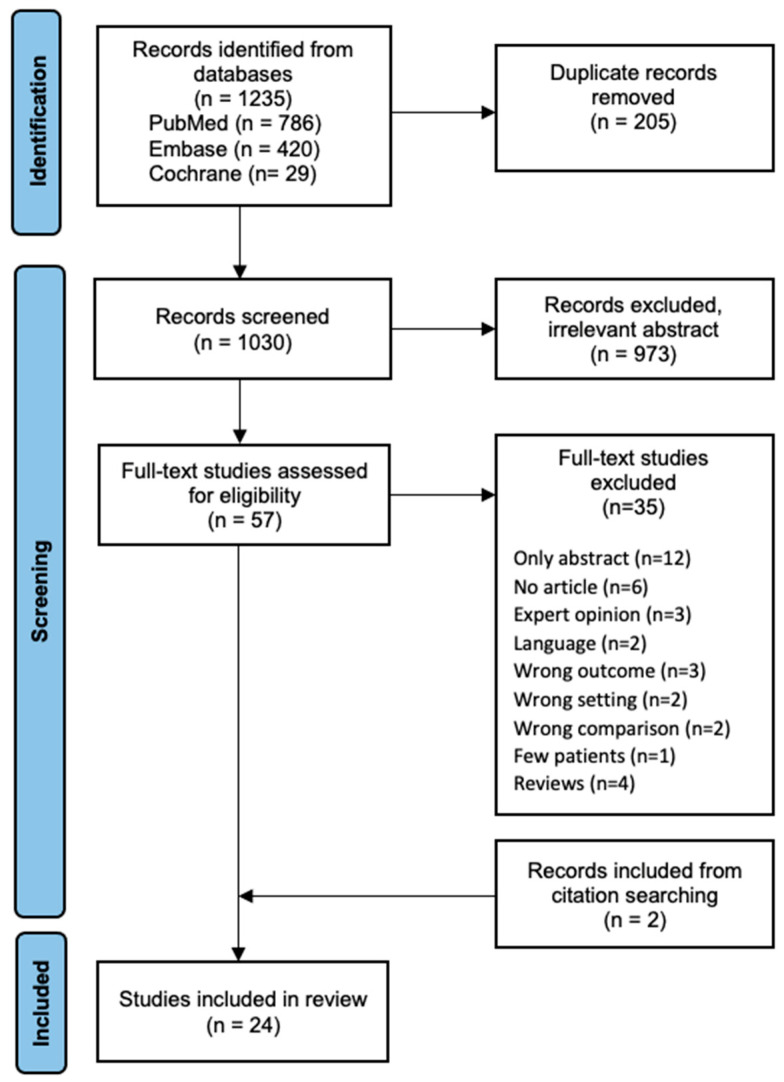
Flow diagram outlining selection of studies included in a systematic review.

**Table 1 jcm-12-02219-t001:** Study characteristics and study outcomes.

Author Year Country	Study Design	Participants/ Cycles	Subjects	Abstinence Period	Outcome Measurements
Dahan et al. [18] 2020 Canada	Prospective	112/-	Infertility patients, male factor	3 h and 3 days	DNA fragmentation Semen parameters (volume, motility, concentration, morphology and total sperm count)
Kabukçu et al. [19] 2020 Turkey	Randomized controlled trial	106/-	Infertility patients	1 and 3 days	Pregnancy rate and miscarriage rate DNA fragmentation Semen parameters (viscosity, volume, motility, morphology and count)
Agarwal et al. [20] 2016 USA	Prospective	7/-	Volunteers, normozoospermic	<2, 2–7 and 9–11 days 1, 2, 5, 7, 9, and 11 days	DNA fragmentation and reactive oxygen species Semen parameters (viscosity, vitality, volume, pH, concentration, morphology, sperm count)
Borges et al. [21] 2018 Brazil	Prospective	818/483	Infertility patients, only male factor	<4 and >4 days 1, 2, 3 and 4 days	Pregnancy rate, miscarriage rate, fertilization rate, implantation rate and embryo rate DNA fragmentation Semen parameters (volume, concentration, motility, morphology, sperm count)
Vahidi et al. [22] 2021 Iran	Prospective	64/-	Infertility patients, with increased DFI	24 h, 3 and 2–7 days.	DNA fragmentation and DNA protamination Semen parameters (volume, concentration, motility, morphology)
Comar et al. [23] 2017 Brazil	Prospective	2458/-	Infertility patients	<2, 2–5 and >5 days	DNA fragmentation, DNA protamination and mitochondrial membrane potential Semen parameters (pH, volume, concentration, motility, normal sperm forms, leukocytes, vitality, apoptosis)
Uppangala et al. [24] 2016 India	Prospective	19/-	Healthy Volunteers	1, 3, 5 and 7 days	DNA fragmentation, sperm chromatin maturity and hypermethylation levelSemen parameters (volume, concentration, motility, morphology, vitality, viability)
Sánchez-Martín et al. [25] 2013 Spain	Prospective	190/-	Infertility patients, non-severe male factor	12 h and 4 days	DNA fragmentation Semen parameters (volume, concentration, motility)
Scarselli et al. [26] 2019 Italy	Prospective	22/265	Infertility patients, OAT	1 h and 2–5 days	Pregnancy rate, fertilization rate, implantation rate and blastocyst rate Semen parameters (volume, concentration, motility, morphology)
Shen et al. [27] 2019 China	Prospective	528/-	Infertility patients	1–3 h and 3–7 days	Pregnancy rate, live birth rate, miscarriage rate and implantation rate DNA fragmentation, acrosome reaction, antioxidant capacity, mitochondrial membrane potential, DNA stainability, nucleoprotein transition and reactive oxygen species Semen parameters (volume, count, concentration, motility, vitality, morphology)
Gosálvez et al. [28] 2011 Spain	Prospective	33/-	(1) Infertility patients with female factor (2) Donors	(1) 24 h and 4 days (2) 3 h and 4 days	DNA fragmentation
Jurema et al. [29] 2005 USA	Retrospective	417/929	Infertility patients, unexplained or oligomenorrhea	≤3, 3–10 and >10 days	Pregnancy rate Semen parameters (concentration, motility, morphology, total motile sperm)
Mayorga-Torres et al. [30] 2015 Colombia	Prospective	6/-	Volunteers, normozoospermic	1 and 3–4 days	DNA fragmentation, mitochondrial membrane potential, membrane integrity, Reactive oxygen species Semen parameters (volume, concentration, motility, morphology, count, vitality)
Jonge et al. [31] 2004 Belgium	Prospective	11/-	Infertility patients	1, 3, 5 and 8 days	DNA fragmentation and DNA stainability Semen parameters (volume, pH, concentration, motility, morphology, viability)
Kulkarni et al. [32] 2022 India	Prospective	67/-	Infertility patients	1–3 h and 2–7 days	DNA fragmentation Semen parameters (volume, count, concentration, motility)
Marshburn et al. [33] 2010 USA	Retrospective	372/866	Infertility patients, normospermia andoligozoospermia	<2, 3–5 and >5 days	Pregnancy rate Semen parameters (volume, concentration, total motile sperm, numbers of dead sperm)
Barbagallo et al. [34] 2021 Italy	Prospective	313/-	Infertility patients, normozoospermic and OA	1 h and 2–7 days	Pregnancy rate, live birth rate, miscarriage rate, fertilization rate, implantation rate, embryo quality, type of birth and birth weight Semen parameters (concentration, motility, morphology)
Kably-Ambe et al. [35] 2015 Mexico	Retrospective	3123/3123	Infertility patients	0–1, 2–3, 4–5, 6–7, 8–9, 10–14 and 15–20 days	Pregnancy rate and recovery rate Semen parameters (volume, concentration, progressive motility, morphology)
Gupta et al. [36] 2021 India	Prospective Retrospective analysis	-/1691	Infertility patients, normozoospermic	1, 2–5, 6–7 and ≥8 days	Pregnancy rate, miscarriage rate, fertilization rate, implantation rate, positive β-hCG rate, embryo sacs and ectopic pregnancy Semen parameters (volume, concentration, motility, morphology)
Manna et al. [37] 2020 Italy	Prospective	65/-	Infertility patients, normozoospermic and OAT	1 h and 2–7 days	DNA fragmentation Semen parameters (volume, concentration, motility, morphology)
Azizi et al. [38] 2021 Iran	Retrospective	1003/1003	Infertility patients, male and female factor	1, 2, 3, 4, 5 and 6–10 days	Pregnancy rate, fertilization rate and cleavage-stage embryo rate Semen parameters (volume, count, concentration, motility, morphology)
Welliver et al. [39] 2016 USA	Prospective	20/-	Normozoospermic	1 and 3–5 days	DNA fragmentation Semen parameters (volume, concentration, motility, pH, total motile count, morphology)
Periyasamy et al. [40] 2017 India	Retrospective	-/1030	Infertility patients, male and female factor	2–7 and >7 days.2–4 and 5–7 days *	Pregnancy rate, live birth rate, miscarriage rate, fertilization rate, implantation rate and cleavage-stage grade embryo rate
Lee et al. [41] 2018 Korea	Retrospective	-/449	Infertility patients, male and female factor	2–7 and 8 days	Pregnancy rate, miscarriage rate and implantation rate

DFI: DNA fragmentation index, OAT: oligoastenoteratozoospermic, OA: obstructive azoospermia, -: not reported, * subgroup of participants undergoing EA of 2–7 days.

## Data Availability

Not applicable.

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
