# Peer review of "The Influence of Male Ejaculatory Abstinence Time on Pregnancy Rate, Live Birth Rate and DNA Fragmentation: A Systematic Review"

_jcm, 2023, doi:10.3390/jcm12062219_

Round 1

Reviewer 1 Report

This is a well-written comprehensive review of the influence of male ejaculatory abstinence. The authors conducted a detailed review of 24 studies aiming to compare short and long ejaculatory abstinence time and investigate its association with fertility outcomes and semen quality. The objective is clearly presented and supported by the systematic review results. The study selection and study results are clearly presented with detailed introduction and discussion sections. The paper adds value to the relative research field and scientific community. I have no additional questions.

Author Response

Thank you so much for your comments. We appreciate your contribution to our work.

Reviewer 2 Report

The systematic review conducted by Freja Sørensen et al. entitled “The influence of male ejaculatory abstinence time on pregnancy rate, live birth rate and DNA fragmentation: A Systematic Review” compared short and long ejaculatory abstinence time and investigated the association to fertility outcomes and semen quality. The main objective of review was pregnancy rate and secondary objectives were live birth rate and DNA fragmentation index. The authors concluded that pregnancy rate, live birth rate and DNA fragmentation was likely to be improved with short ejaculatory abstinence compared to long ejaculatory abstinence. However, the authors state that it is not possible conduct a clear recommendation on the ideal timeframe for ejaculatory abstinence due to heterogenous abstinence times, so this systematic review cautiously recommends an ejaculatory abstinence of one day to improve pregnancy and live birth rate and decreased level of DNA fragmentation in semen followed by ART. I think that a review is well done also emphasizing the strengths and limits of the study conducted.  Please study in deep the discussion. Please formatting needs to be revised.

Author Response

Thank you very much for your comments. We appreciate your contribution to our work. We have revised sections of the discussion and hope you find the changes adequate. The manuscript has been edited by a skilled English speaker, and we hope that you find the corrections made to be satisfactory.

Reviewer 3 Report

The titled manuscript is a systematic review that shows that ejaculatory abstinence can be with shorter intervals than described by WHO, since it improves reproductive rates and reduces DNA fragmentation. In general, the manuscript is correctly written, easy to read and very interesting, and apparently a topic that is becoming very relevant in recent years, surely the creation of scientific evidence will allow entities such as WHO to update their recommendations, particularly in those couples that require some reproductive management. I only leave some minor comments

Line 44: include the meaning of WHO, you do it on line 61

Table 2: the meaning of some acronyms such as ICSI, IVF and others are missing

Table 2: I don't understand why EA are presented as intervals days and successive days also appear (Agarwal and Borges). EA of Vahidi study says 24 h, 3 d and 2-7 d. Day 3 would not be included in the interval 2-7?. The EA of Periyasamy study, is not understood either

Figure 2: this paragraph “Solid line: significantly different (p<0.05), dotted line: not significantly different (p>0.05), EA: ejaculatory abstinence time” It should be part of the figure caption. you should include the reference number, as you did in the tables.

Section 3.3: There is no longer line numbering. What do you mean by frozen thawed cycles?

Section 3.5: I think this section should be immediately after the item selection of the studies (3.1)

Thanks for invite me to review this manuscript

Author Response

“The titled manuscript is a systematic review that shows that ejaculatory abstinence can be with shorter intervals than described by WHO, since it improves reproductive rates and reduces DNA fragmentation. In general, the manuscript is correctly written, easy to read and very interesting, and apparently a topic that is becoming very relevant in recent years, surely the creation of scientific evidence will allow entities such as WHO to update their recommendations, particularly in those couples that require some reproductive management. I only leave some minor comments
Thank you very much for your comments. We appreciate your contribution to our work.

Line 44: include the meaning of WHO, you do it on line 61

We have now included the meaning of WHO the first time it is mentioned in line 44.

Table 2: the meaning of some acronyms such as ICSI, IVF and others are missing

The acronyms are now explained, both in Table 3 and in the text.

Table 2: I don't understand why EA are presented as intervals days and successive days also appear (Agarwal and Borges). EA of Vahidi study says 24 h, 3 d and 2-7 d. Day 3 would not be included in the interval 2-7?. The EA of Periyasamy study, is not understood either.

We explained the interval and successive EAs of the studies in line 207-210 and for Periyasamys EA, please see Table 1 Asterix (*)

Figure 2: this paragraph “Solid line: significantly different (p<0.05), dotted line: not significantly different (p>0.05), EA: ejaculatory abstinence time” It should be part of the figure caption. you should include the reference number, as you did in the tables.

We moved the explanatory text and included reference numbers, please see line 223-225

Section 3.3: There is no longer line numbering. What do you mean by frozen thawed cycles?

Please see line 198-201, “frozen thawed cycles” refers to the process of using previously cryopreserved blastocysts in fertility treatment. This is now explained in the text.

Section 3.5: I think this section should be immediately after the item selection of the studies (3.1)”

We have now moved this section in accordance with this comment.

Reviewer 4 Report

In this systematic review, 24 studies were selected through comprehensive search strategy detection. Through comprehensive analysis of 24 studies, it was found that compared with long EA, short EA might improve pregnancy rate, live birth rate and DNA fragmentation. This systematic review basically includes the steps of systematic evaluation, but it seems to lack the methodological quality analysis of the selected research.

Author Response

Thank you very much for your comments. We appreciate your contribution to our work. We have included a comment on the methodological questions according to SIGN and explained why some of those were not applicable to our study, please see line 130-133. Our systematic review has been conducted in accordance with the current PRISMA guideline. We had hoped to conduct more extensive analyses on the studies, including a meta-analysis. However, the studies were not able to support such analyses due to heterogeneity. As a result, we conducted data analyses that were within the capacity of the studies, in order to ensure that our conclusions were supported by the available evidence. We hope that you will find that the studies are able to support the conclusions that we draw given the revisions that we have made, particularly in the discussion and conclusion sections.

Reviewer 5 Report

The MS is a systematic review about the influence of abstince time (AT) on reproductive outcomes. There are already recent review on this topic and the Authors should consider this fact. In particular, one is focused on DNA fragmenation (Barbagallo F, et al The Impact of a Very Short Abstinence Period on Conventional Sperm Parameters and Sperm DNA Fragmentation: A Systematic Review and Meta-Analysis. J Clin Med. 2022 Dec 8;11(24):7303. ) and is published in the same journal. Others are more focused on the effect of AT on semen parameters. Authors should consider overlap, although there is less (in terms of review papers) on pregnancy rate and LBR. .

specific points:

page 2 line 52 During transit... Please consider to change this sentence: not always spermatozoa are exposed to ROS during transit, they may be exposed especially when an inflammatory condition is present.

page 2 lines 61-73. I think that here the Authors should clarify several things regarding the WHO manual. The sixth edition of the WHO manual for semen analysis (it is not a guideline) reccomends 2-7 days of abstinence only for diagnostic purposes. WHO manual has been created only for this aim. The main reason for such reccomandation lays in the fact that diagnostic laboratories should have standard for the analysis, and this standard must be followed especially when the laboratory, in the report for the patient, use the reference table that is present in the manual VI edition (appendix). The values reported in the table refer to men who had an AT between 2 and 7 days, so such reference table can be use donly if this AT is respected and if standard procedures as described in the manual, are followed. So the paragraph should be changed, because WHO manual does not reccomend 2-7 days AT for other purposes (such for instance semen collection for IVF or ICSI), but only for diagnosis.

regarding the results, the AUthors should consider whether some of the studies on sperm DNA fragmentation have been performed by comparing AT in the same patient. If such studies exist the Authors should add a paragraph commenting them (for instance in the discussion).   studies  on pregnancy rate and live birth rate compare different couples. From the report, it appears that in some studies the caseloads are low. However, this is a systematic review not a metanalysis, so the Authors did a correct analysis and considered the biases of the studies.  

in the discussion, (page 12), the AUthors discuss about the inclusion in their review of retrospective studies which could have in creased the risk of selection bias. I think that in a systematic review Authors should include all the studies they found and, of course, criticize them if needed. So I would not say that inclusion of these studies is a limitation of the paper but that overall, such studies may not be useful or of little help to reach a clear-cut conclusion.  Again this is not a metanalysis.

page 12: the sentence: according to WHO reccomendation should be removed (see above: WHO do not reccomend such AT fo ART).

Overall, this review is simply confirmatory of other metanalisis (which include, as written here, much less stdies, but in a metanalysis only high quality studies should be included in order to reach a conclusion for clinical evidence).

The sentence (page 13) This systematic review reccomends revision of the WHO......... a day should be eliminated for the reasons outlined above.

Overall, this review cannot reccomend anything from a clinical point of view because the evidence is not so strong and this is not a metanalysis, so I would avoid the word "reccomends" (even if cautiosly) in the conclusion. Maybe te Authors may add somenthing in the discussion regarding those patients who have low semen quality and high DNA fragmenation levels due to inflammatory problems in the male genital tract. In these patients a lower AT might be of help.

Author Response

The MS is a systematic review about the influence of abstince time (AT) on reproductive outcomes. There are already recent review on this topic and the Authors should consider this fact. In particular, one is focused on DNA fragmenation (Barbagallo F, et al The Impact of a Very Short Abstinence Period on Conventional Sperm Parameters and Sperm DNA Fragmentation: A Systematic Review and Meta-Analysis. J Clin Med. 2022 Dec 8;11(24):7303. ) and is published in the same journal. Others are more focused on the effect of AT on semen parameters. Authors should consider overlap, although there is less (in terms of review papers) on pregnancy rate and LBR.

specific points:

Thank you very much for your comments. We appreciate your contribution to our work. We have included a comment in our manuscript regarding the new systematic review from 2022, please see line 292-296. The study conducted a meta-analysis on four studies related to DNA-fragmentation, which we believe is not directly overlapping with our study, which primarily focus on pregnancy rate.

page 2 line 52 During transit... Please consider to change this sentence: not always spermatozoa are exposed to ROS during transit, they may be exposed especially when an inflammatory condition is present.
The sentence has been revised according to this comment, see line 53.

page 2 lines 61-73. I think that here the Authors should clarify several things regarding the WHO manual. The sixth edition of the WHO manual for semen analysis (it is not a guideline) reccomends 2-7 days of abstinence only for diagnostic purposes. WHO manual has been created only for this aim. The main reason for such reccomandation lays in the fact that diagnostic laboratories should have standard for the analysis, and this standard must be followed especially when the laboratory, in the report for the patient, use the reference table that is present in the manual VI edition (appendix). The values reported in the table refer to men who had an AT between 2 and 7 days, so such reference table can be use donly if this AT is respected and if standard procedures as described in the manual, are followed. So the paragraph should be changed, because WHO manual does not reccomend 2-7 days AT for other purposes (such for instance semen collection for IVF or ICSI), but only for diagnosis.

We have changed the word guideline to manual in line 62. We have additionally elaborated that the WHO recommendations are aimed at semen collection, line 62-67. We understand abovementioned concerns regarding a revision of WHO recommendations. Therefore, we have revised the discussion section and aimed our recommendation for fertility treatment and not diagnostic purposes, please see line 306-315.

regarding the results, the Authors should consider whether some of the studies on sperm DNA fragmentation have been performed by comparing AT in the same patient. If such studies exist the Authors should add a paragraph commenting them (for instance in the discussion). studies on pregnancy rate and live birth rate compare different couples. From the report, it appears that in some studies the caseloads are low. However, this is a systematic Review not a metanalysis, so the Authors did a correct analysis and considered the biases of the studies.  
Information about studies who did not used the same patient as control is now mentioned, please see line 210-214.

in the discussion, (page 12), the Authors discuss about the inclusion in their review of retrospective studies which could have in creased the risk of selection bias. I think that in a systematic review Authors should include all the studies they found and, of course, criticize them if needed. So I would not say that inclusion of these studies is a limitation of the paper but that overall, such studies may not be useful or of little help to reach a clear-cut conclusion.  Again this is not a metanalysis.

We have now removed the sentence that stated that the inclusion of retrospective studies was a weakness of our study. 

page 12: the sentence: according to WHO reccomendation should be removed (see above: WHO do not reccomend such AT fo ART).

We have now removed the sentence in line 261 but it should be noted that the systematic review we are referring to (reference 13), formed their EA groups in accordance with WHO recommendations. Therefore, the group with EA of 2-7 day was referred to as “RAP = WHO recommended abstinence period”. Therefore, recommendations set by WHO may be misunderstood when initiating fertility treatment.

Overall, this review is simply confirmatory of other metanalisis (which include, as written here, much less stdies, but in a metanalysis only high quality studies should be included in order to reach a conclusion for clinical evidence). 

We respectfully disagree with this comment. This systematic review including 24 studies also investigates other outcomes of ejaculatory abstinence times other than supporting the existing meta-analysis which only focusses on pregnancy rates.

The sentence (page 13) This systematic review reccomends revision of the WHO......... a day should be eliminated for the reasons outlined above.
We have now made it clear that WHO only recommends 2-7 days of EA for diagnostic purpose, please see line 308-310.

Overall, this review cannot reccomend anything from a clinical point of view because the evidence is not so strong and this is not a metanalysis, so I would avoid the word "reccomends" (even if cautiosly) in the conclusion. Maybe te Authors may add somenthing in the discussion regarding those patients who have low semen quality and high DNA fragmenation levels due to inflammatory problems in the male genital tract. In these patients a lower AT might be of help.

We have now taken out the word “recommends” from the conclusion and instead emphasized that there is a tendency in the discussion section, please see line 317-321.

Round 2

Reviewer 4 Report

In this systematic review, 24 studies were selected through comprehensive search strategy detection. Through comprehensive analysis of 24 studies, it was found that compared with long EA, short EA might improve pregnancy rate, live birth rate and DNA fragmentation. 

Author Response

Thank you for taking the time to leave a comment. We appreciate your feedback and are pleased that you found our recent revisions satisfactory.

Reviewer 5 Report

The Authors answered to my comments and amended the MS. There are still some corrections to be done:

1. In the abstrac the sentences: WHO (World Health Organization) recommend 2-7 days of abstinence time prior to semen collection, however the evidence for this recommendation remainsis unclear. This sentence should be changed considering my comments: WHO (World Health Organization) recommend 2-7 days of abstinence time prior to semen collection for diagnostic purposes, however the evidence that such abstinence period gives the better semen quality for assisted reproduction purposes remains unclear. Also, the sentence: "  This systematic review confirms that short ejaculatory abstinence time, less than recommended by
WHO, are associated with higher pregnancy and live birth rates and improved DNA fragmentation, when compared to long ejaculatory abstinence time"
should be changed in:This systematic review confirms that short ejaculatory abstinence time, less than recommended by WHO for diagnostic purposes, are associated with higher pregnancy and live birth rates and improved DNA fragmentation, when compared to long ejaculatoryer abstinence time

Introduction: the sentence: "This recommendation has remained unchanged since their first edition in 1980, however the evidence supporting these recommendations are unclear [12]. " should be deleted, in view of my previous comments (see report 1). Reccomandation of WHO is referred to the use of reference limits or the reference table (last edition), so the sentence is oncorrect.

The sentence: In contrast to the recommendations, several recent studies have reported a correlation between short EA
and lower DNA fragmentation and thus better semen quality [13]" In contrast to the reccomendation must be deleted .

the sentence: "Also, the European Society of Human Reproduction and Embryology (ESHRE) is not supporting the WHO recommendation and is currently recommending a shorter and narrower range from 3-4 days of EA" should be changed in: "Also, the European Society of Human Reproduction and Embryology (ESHRE)  is currently recommending a
shorter and narrower range from 3-4 days of EA"

the discussion is ok

.

Author Response

The Authors answered to my comments and amended the MS. There are still some corrections to be done:

Thank you for your thorough feedback on our work. We appreciate the time you took to review our revisions and have made the necessary adjustments to meet your expectations. We hope you find the updated version satisfactory.  

  1. In the abstract the sentences:WHO (WorldHealth Organization) recommend 2-7 days of abstinence time prior to semen collection, however the evidence for this recommendation remains unclear. This sentence should be changed considering my comments: WHO (World Health Organization) recommend 2-7 days of abstinence time prior to semen collection for diagnostic purposes, however the evidence that such abstinence period gives the better semen quality for assisted reproduction purposes remains unclear.
    We have changed the sentence, please see line 18-20.

    Also, the sentence: "  This systematic review confirms that short ejaculatory abstinence time, less than recommended by WHO, are associated with higher pregnancy and live birth rates and improved DNA fragmentation, when compared to long ejaculatory abstinence time" should be changed in: This systematic review confirms that short ejaculatory abstinence time, less than recommended by WHO for diagnostic purposes, are associated with higher pregnancy and live birth rates and improved DNA fragmentation, when compared to long ejaculatory abstinence time.
    We have changed the sentence, please see line 36.
  2. Introduction:
    the sentence: "This recommendation has remained unchanged since theirfirst edition in 1980, however the evidence supporting these recommendationsare unclear [12]. "
    should be deleted, in view of my previous comments (see report 1). Reccomandation of WHO is referred to the use of reference limits or the reference table (last edition), so the sentence is incorrect. 
    We have deleted the sentence.

    The sentence: In contrast to the recommendations, several recent studies have reported a correlation between short EA and lower DNA fragmentation and thus better semen quality [13]" In contrast to the reccomendation must be deleted.
    We have changed the sentence, please see line 65.

the sentence: "Also, the European Society of Human Reproduction and Embryology (ESHRE) is not supporting the WHO recommendation and is currently recommending a shorter and narrower range from 3-4 days of EA" should be changed in: "Also, the European Society of Human Reproduction and Embryology (ESHRE) is currently recommending a 
shorter and narrower range from 3-4 days of EA"
We have changed the sentence, please see line 70-72.

the discussion is ok